# Willingness of the Jordanian Population to Receive a COVID-19 Booster Dose: A Cross-Sectional Study

**DOI:** 10.3390/vaccines10030410

**Published:** 2022-03-09

**Authors:** Walid Al-Qerem, Abdel Qader Al Bawab, Alaa Hammad, Jonathan Ling, Fawaz Alasmari

**Affiliations:** 1Faculty of Pharmacy, Al-Zaytoonah University of Jordan, Amman 11733, Jordan; abdelqader.albawab@zuj.edu.jo (A.Q.A.B.); alaa.hammad@zuj.edu.jo (A.H.); 2Faculty of Health Sciences and Wellbeing, University of Sunderland, Chester Road, Sunderland SR1 3SD, UK; jonathan.ling@sunderland.ac.uk; 3Department of Pharmacology and Toxicology, College of Pharmacy, King Saud University, Riyadh 4545, Saudi Arabia; ffalasmari@KSU.EDU.SA

**Keywords:** COVID-19, booster dose, hesitancy, Jordan

## Abstract

SARS-CoV-2 (COVID-19) vaccines are critical for containing serious infections. However, as COVID-19 evolves toward more transmissible varieties and serum antibody levels in vaccinated persons steadily decline over time, the likelihood of breakthrough infections increases. This is a cross-sectional study based on an online questionnaire for Jordanian adults (n = 915) to determine how individuals who have finished the current vaccination regimen feel about a prospective COVID-19 booster shot and what factors might influence their decision. Almost half of the participants (44.6%) intended to get the booster dose of the COVID-19 vaccine. The most frequently mentioned reasons for participants’ reluctance to get the COVID-19 vaccine booster dose were “The benefits of booster dose have not been scientifically proven” (39.8%), followed by “I took the last dose a short time ago, and there will be no need to take the booster dose for at least a year” (24.6%). In turn, “I was infected with COVID-19; thus, I do not require the booster dose” was the least reported reason (13.1%). These findings highlight the considerable hesitancy toward COVID-19 booster immunization among Jordanians, as well as the variables associated with vaccine hesitancy and the most frequently mentioned reasons for hesitancy, which will aid in creating excellent campaigns regarding booster doses.

## 1. Introduction

Vaccines are one of the most efficient medical interventions for the control of infectious diseases [1]. The emergence of the COVID-19 pandemic, has caused over 5 million deaths globally and 11,197 deaths in Jordan as of 13 November 2021 [2]. Extensive research has been conducted to develop a safe and effective vaccine to control the spread of the virus. The vaccine development process has had huge financial and institutional support, which aided in an unprecedentedly accelerated development of hundreds of vaccine candidates for pre-clinical and clinical studies. As of September 2020, four COVID-19 vaccines had emergency use authorization by the WHO following phase III clinical trials, a process that normally takes decades. Recent data indicate that there are 18 COVID-19 vaccines approved and currently used worldwide [3,4].

Authorized vaccines have been found to be highly effective in reducing hospitalization and moderate to severe disease, particularly in fully vaccinated individuals who have risk factors for developing complications such as old age, obesity, chronic diseases, and immunosuppression [5,6]. With the emergence of new SARS-CoV-2 variants, dominated by the highly transmissible Delta (B.1.617.2) variant, the efficacy of vaccines against infection with this variant and other so-called “variants of concern” was found to be reduced [7]. This highlighted the need for a third dose that is active against new variants. In addition, cumulative data found that a reduction in serum anti-spike IgG antibody levels occurs several months after vaccination, creating a need for a booster dose of the vaccine, especially for individuals at high risk of severe infection [4].

On 30 July 2021, Israeli authorities authorized the administration of a booster. Participants aged 60 and over, as well as those who had had a booster at least 5 months after receiving a second dose of BNT162b2, had a 90% lower death rate due to COVID-19 than those who did not get a booster [8]. As of 24 December 2021, 477.27 million booster vaccinations had been carried out worldwide (6.06 doses per 100 people), particularly China, the United States, the United Kingdom, and Israel. This has enhanced vaccination to strengthen local population immunity, raised resilience to variations, and maintained social order stability [9]. The total acceptability percentage for a booster vaccine among healthcare professionals in the United States was 83.6% [10]. Furthermore, it was strong (71.3%) among Czech healthcare professionals, with 12.2% uncertain and 16.6% against [11]. In Poland, 71% of adults expressed willingness to accept the booster dose [12]. In 2021, Sugawara et al. reported that the booster vaccine was viewed positively by Japanese medical students (89.1%) [13]

However, the success of the COVID-19 booster immunization programs is dependent on patients’ willingness to receive the booster dose. Because of the serious side effects reported for COVID-19 vaccinations, particularly through social media [14], people’s hesitancy toward taking additional doses of the vaccine has increased. This is becoming a significant barrier in the immunization process and also increasing outbreaks of vaccine-preventable illnesses [12,15].

Influenza remains the best well-known model among the vaccine-preventable epidemic/pandemic airborne diseases prior to COVID-19. Therefore, influenza vaccination example can be used to understand the phenomenon of vaccine hesitancy. Several studies have investigated the reasons behind vaccine hesitancy regarding the influenza vaccine in different populations, especially healthcare professionals. These studies have discovered that the main determinants of hesitation are (1) insufficient awareness campaigns; (2) altered risk perception; (3) insufficient health education on the efficacy of the influenza vaccine and/or potential adverse reactions; (4) lack of access to vaccination facilities; and (5) socio-demographic variables. Furthermore, according to various studies, one of the key factors of low flu vaccine uptake among healthcare workers is a lack of time to attend vaccination clinics [16]. Therefore, one of the suggested methods to combat vaccination hesitancy is offering an integrated vaccination complex [16].

A population’s decision to get vaccinated is embedded in a specific social environment of ideas and perceptions, as well as concerns related to vaccine availability and costs [17]. The notion of vaccine hesitancy was proposed by the World Health Organization’s Strategic Advisory Group of Experts with the goal of evaluating the social variables that lead to a delay in vaccine acceptance or rejection despite the availability of immunization services [18]. The issue of vaccine hesitancy has increased with the emergence of social media that has increased the spread of misconceptions associated with vaccination [19].

The information about Jordanian acceptance of a booster vaccine dose against COVID-19 is lacking. Therefore, the aim of the current study is to assess the willingness of the Jordanian population who were previously vaccinated with two doses toward receiving a booster dose.

## 2. Materials and Methods

This is a cross-sectional study based on an online questionnaire created using Google Forms. The link was shared with several Jordanian all-purpose Facebook groups. Questions on age and place of residence were included in the questionnaire to ensure that individuals met the inclusion requirements of being aged 18 or above and living in Jordan. Importantly, only participants who had both the first and the second dose of the vaccine (complete vaccine course) were included in the study. The data were collected between 1 October and 15 December 2021. In that period, when the data were collected, the pandemic in Jordan rose from 700 daily new confirmed cases on the 1 October to 5000 daily new confirmed cases by 15 December [20]. At the beginning of data collection more than 3,290,000 adults had been completely vaccinated [21], which represents around 49% of the adult population of Jordan [22]. This study was implemented according to the principles of the Declaration of Helsinki and granted ethical approval by the Al-Zaytoonah Research Ethics Committee.

### 2.1. Sampling Type and Sample Size

In this research, convenience sampling was applied. The sample size was calculated based on a 95% significance level and a 4% margin of error. The estimated sample size was 601 and the study had 915 individuals, indicating that the required sample size was met. To compute the minimal required sample size to perform a multinominal logistic regression, the authors adopted the rule of Events Per Variable criterion (EPV) ≥ 10 [23]. The dependent variable (“Are you willing to take the booster dose?”) had three levels: “Yes”, “No”, and “Not sure”. The smallest group in the three possible outcomes was “Not sure”, which included 226 participants. Thus, the number of independent variables that could have been included in the model was 22. The model used in the current study included only 12 variables.

### 2.2. Survey Validity and Reliability

The survey was originally written in English, after conducting a comprehensive literature research. Following this, the questionnaire was translated into Arabic and back to English by various translators, with the two English versions being judged to be comparable. An expert panel evaluated the generated survey’s content validity. The questionnaire was emailed to 30 people to confirm the face validity, and revisions were made based on their feedback. These participants’ data were not included in the final analysis. Cronbach’s alpha was used to assess the reliability of the generated questionnaire scores, which included general information about COVID-19 and COVID-19 vaccines, practices towards COVID-19, as well as knowledge about the third vaccine dose. Because a lower Cronbach’s alpha is expected with binary data [24] and a small number of questions [25], the acceptable Cronbach’s alpha range was judged to be above 0.5 [26].

### 2.3. Study Instrument and Data Collection

The questionnaire was adapted from previously formulated questionnaires [27,28]. A prologue that describes the aims and contexts of the study and contains an informed consent form were included at the beginning of the questionnaire. The survey starts with a question about receiving the complete vaccination course against COVID-19. If the participant was not completely vaccinated, they were thanked for their interest in the study and completed no further questions.

The questionnaire consisted of six branched sections (Appendix A). The first part of the survey gathered the socio-demographic data of the participants, such as gender, marital status, pregnancy status, and whether they have children, as well as their smoking habits, weight status, health status, education level, and average household monthly income (measured in Jordanian dinars, JOD). The second section included questions about individuals’ perception of COVID-19. For example, one question asked the respondents to assess the COVID-19 disease severity on a scale of 1 (not serious) to 5 (extremely serious), while another question asked if the participants had been infected with COVID-19. The third segment collected data on the participants’ knowledge of COVID-19, practices to protect against COVID-19 infection, and mode of transmission. The fourth part investigated the type of vaccine received by the participants and any side effects they had experienced from it. The fifth section measured their knowledge of the vaccine third (booster) dose. Finally, the sixth component looked at the factors that influence participants’ acceptance of the third dose of vaccine. Health status information collected by the questionnaire were used to categorize the participants based on the Centers for Disease Control and Prevention (CDC) criteria of low-, medium-, and high-risk groups to develop complications due to COVID-19 [28,29].

In this research, the knowledge and practice scores were calculated separately. The maximum possible score for knowledge was 21, with one point awarded for each correct answer (the score was calculated based on items in Appendix A). Based on their knowledge scores, the participants were separated into high vs. low knowledge groups.

As for the practice score, the score was calculated as the sum of the practice statements for the question “What procedures have you used to protect yourself from COVID-19?” (Appendix A), which represent the participants’ adherence to protective measures with answers ranging from “Never” (1 point) to “All the time” (5 points), with a maximum possible score of 25.

The medians for the knowledge and practice scores were calculated and the participants were divided into two groups; the first group (high-level group) included participants who had higher than the median scores and the rest were included in the low-level group.

### 2.4. Statistical Analysis

Categorical variables were presented as frequencies and percentages, and continuous variables were presented as the means and standard deviations (SD). Univariate analysis using Kruskal–Wallis and Chi-square analyses were used to identify variable associations with participants’ intention to receive the booster dose. A multinominal logistic regression with participants’ intention to receive the booster dose as the dependent variable was conducted. The independent variables included in the model were the participants’ age, material status, having children, education level, household average monthly income, severity of their COVID-19 vaccine-related side effects, reasons behind taking the first doses of the vaccine, knowing someone who had died due to COVID-19, knowledge level, adherence to protective measures against COVID-19, and perceived risk of developing COVID-19 complications. Due to the high correlations between vaccine type and severity of side effects reported by the participants, vaccine type was excluded from the analysis. Kruskal–Wallis was applied to evaluate the differences in side effects severity scores by different vaccination type groups. IBM SPSS version 27 was used to analyze the data.

## 3. Results

### 3.1. Sample Charecteristics

Sample characteristics are presented in Table 1. Nine hundred and fifteen participants were enrolled in the study. The majority were between 18 and 29 years (45.7%), and half (49.9%) were married. Only (22.2%) of the participants had a chronic disease. Almost half (47.3%) had a child, 72.1% did not smoke, and 69% were living in Amman, the capital city. Most of the participants were bachelor’s degree holders (64%). Forty-three percent of the participants earned 500–1000 JD per month and 35.3% of the participants experienced moderate adverse effects with the COVID-19 vaccine. Almost three-quarters (74%) knew someone who had died due to COVID-19. Finally, the most reported side effect experienced due to the COVID-19 vaccine was pain at the injection site (67.2%).

### 3.2. Sample Charecteristics’ Association with Booster Dose Acceptance

Protective practices against COVID-19 and knowledge levels were computed for each participant (Appendix A). The medians were 23 (out of maximum possible score of 25) and 15 (out of maximum possible score of 21) for practices and knowledge, respectively. Cronbach’s alpha indicated good internal consistency for the knowledge and practice scores (0.63 and 0.8, respectively). As Table 2 shows, only 408 (44.6%) of the participants in the current sample intended to get the booster dose of the COVID-19 vaccine, while the rest were either “not sure” 226 (24.7%) or intended not to receive the booster dose of the COVID-19 vaccine 281 (30.7%). Univariate analysis was conducted using Chi-square and Kruskal–Wallis tests (Table 2). Results showed that several variables were associated with intention to get the booster dose, including perceived side effects of the vaccine (*p* = 0.03), participants taking the COVID-19 vaccine out of conviction, or because of enforcing laws (*p* < 0.001), perceived seriousness of COVID-19 (*p* = 0.03), risk level (*p* = 0.009), and vaccine type (*p* < 0.001).

Multinomial regression was applied to evaluate the factors associated with responding “Not Sure” or “No” regarding the intent to get the booster dose of the COVID-19 vaccine (Table 3). The participants who earned 500–1000 JD per month increased the odds of responding “Not sure” vs. “Yes”, with an OR of 1.79 (CI 1.07–3.0).

Participants who experienced mild symptoms or no symptom from the COVID-19 vaccine had lower odds of responding “No” vs. “Yes” (OR 0.54 (CI 0.31–0.97) and OR 0.34 (CI 0.18–0.65), respectively). Those who received the COVID-19 vaccine because of imposed laws, rather than because they believed in it, were 21 times more likely to respond “No” vs. “Yes”, with an OR of 20.88 (CI 13.13–33.21), while it increased the odds seven-fold to respond “Not sure” vs. “Yes”, with an OR of 7.42 (4.58–12.01), and for those who took it out of conviction and because of the laws increased the odds of responding “No” or “Not sure” vs. “Yes” (OR 2.68 (CI 1.66–4.34) and OR 4.99 (CI 3.27–7.63), respectively). Participants who were classified into the low-risk group of developing COVID-19 complications were more likely to answer “Not sure” vs. “Yes”, with an OR of 1.70 (CI 1.06–2.74).

### 3.3. Resported COVID-19 Side Effects by Vaccine Type

There were significant differences between the severity of side effects reported by vaccine type (Figure 1). The most reported percentage of severe/moderate side-effects were reported for the AstraZeneca vaccine followed by Pfizer, while the least reported severe/moderate side effects were reported for Sinopharm.

### 3.4. Reported Reasons for Booster Dose Hesitency/Refusal

Reasons for booster dose hesitancy/refusal are shown in Table 4. The most frequently mentioned causes for participants’ responses regarding their hesitancy to get the booster dose of the COVID-19 vaccine were “The benefits of booster dose have not been scientifically proven” (39.8%), followed by “I took the last dose a short time ago, and there will be no need to take the booster dose for at least a year” (24.6%). In turn, the least mentioned cause was “I was infected with COVID-19; therefore, I do not need the booster dose.” (13.1%).

## 4. Discussion

The main objective of the present research was to examine the willingness of Jordanians, who already received the complete course of the COVID-19 vaccine, to receive a third booster vaccine dose. Another key task was to identify significant predictors associated with the intention to receive a booster dose. The questionnaire utilized was constructed and validated to cover the key aspects the COVID-19 third-dose vaccination. The first section of the questionnaire covered the demographics of the participants. Prior work has found that demographic differences influence beliefs and acceptance of COVID-19 vaccination [30]. The second part of the questionnaire explored the perceived seriousness of COVID-19 [31,32]. The third domain of the questionnaire focused on knowledge and practice related to COVID-19. The fourth domain identified the type of the received vaccine and self-reported vaccine-related side effects. The fifth evaluated the knowledge of the booster dose and the final part evaluated reasons for hesitancy to receive the booster dose. By covering these features related to the terms of COVID-19 vaccination, this study will provide valuable information regarding the prediction of the perceived third-dose vaccine intention/acceptance in Jordan.

The questionnaire was systematically validated to ensure the content and face validity to confirm the appropriateness of using the questionnaire in the current study [33]. Content validity was assessed by personnel with expertise related to COVID-19 pathogenesis and therapeutics confirmed the suitability of the questionnaire items. Face validity was established by a pilot study, which confirmed that the questionnaire items were clear and easy to complete. The forward–backward translation of the questionnaire ensured that it preserved its scientific concepts.

Univariate analysis [34] was used to categorize participants based on how confident they were about having a third booster COVID-19 vaccine shot. Participants’ age, having children, marital status, education, household monthly income, perceived symptoms of the COVID-19 vaccine, personal beliefs about the vaccine, fatality experience of COVID-19, perceived seriousness of COVID-19, risk level for COVID-19 complications, knowledge and protective practice levels regarding COVID-19, and the type of the received vaccine were incorporated into the univariate analysis model. This analysis revealed that taking the COVID-19 vaccine out of conviction and the type of vaccine received were strong predictors for receiving a third booster vaccination. Other factors that were associated with booster dose included perceived side effects of the COVID-19 vaccine, perceived seriousness of COVID-19, and high-risk category of developing COVID-19 complications.

Seventy percent of participants who received the first two COVID-19 vaccine shots out of conviction were more positive about receiving a third booster dose. These findings were concordant to some extent with the literature. A study conducted in China based on the heath belief model found participants showed a high willingness to accept the third dose of the COVID-19 vaccines [35].

We found differences in the likelihood of receiving a booster dose related to the vaccine already received. Of the participants who received Sinopharm, 53.2% reported they would receive a third booster dose compared to 39.6% and 37% of receivers of the Pfizer and AstraZeneca vaccines, respectively. This finding could be attributed to the reported safety of the traditionally manufactured Sinopharm vaccine compared to the DNA- and mRNA-based technology of the Pfizer and AstraZeneca vaccines [36,37]. Furthermore, participants who received the Sinopharm vaccine in the present study reported the fewest side effects. This finding might also contribute to the high acceptance rate for a third booster vaccine dose in this group. Nonetheless, only the level of side effects experienced was incorporated in the regression model. Vaccine type was excluded from the regression model due to multicollinearity, as the type of vaccine and reported severity of side effects were highly correlated.

Of participants who experienced no symptoms after receiving the vaccine, 52.3% stated they would be prepared to receive a third booster COVID-19 vaccination. This compared to 41.2% and 35.3% of respondents who experienced moderate and severe symptoms, respectively, of COVID-19 vaccination. A recent report concluded that the main reasons against accepting a booster COVID-19 dose included the side effects experienced after previous doses and safety uncertainties [3].

The perceived seriousness of COVID-19 influenced decisions to receive a third booster dose. As the score of seriousness increases, participants showed more enthusiasm to receive a third booster COVID-19 vaccination.

Fifty percent of participants who were categorized in the high-level group of developing COVID-19 complications reported that they would receive a third booster dose compared to 38.5% from the lower risk group. A recent report also concluded that patients at higher risk of COVID-19 complications were more hesitant to receive a third booster COVID-19 vaccination [35].

Multinomial logistic regression was used to identify the associated factors that might potentially affect decisions to receiving a third booster COVID-19 vaccination. To a high statistical significance, those who expressed no symptoms with prior COVID-19 vaccinations decreased the odds of rejection of a third booster COVID-19 dose. Participants who were legally required to be vaccinated, rather than being vaccinated because of their beliefs, were more resistant or hesitant to receive a third booster vaccine dose.

The concept of “concordance” describes how patient agreement to receive a particular medication is expected to improve the experience with this medication, hence improving the acceptance of the medication [38]. Exploration of patients’ beliefs can explain medication use and identify barriers that can be resolved to improve medication usage [39]. Interestingly, participants who received the previous vaccine doses both out of conviction and because of the enforcing laws also expressed a high level of resistance and/or hesitancy to receive a third booster COVID-19 vaccine, when compared with the groups who received it solely out of conviction. Nevertheless, we cannot identify how much of the decision making was based on conviction, if any, for those in the conviction and legally obligated combined group.

Accepting the previous doses of the vaccine was not necessarily related to the acceptance of receiving a third vaccine dose. This could be attributed to the experiences related to side-effects from the vaccine rather than accepting or receiving the first doses. For example, a participant may voluntarily accept the first dose of the COVID-19 vaccine, believing that the benefits of the vaccine outweigh the risk of taking it; however, the later experience after vaccination, such as having side effects, may alter beliefs and make people more hesitant for subsequent vaccinations. Previous studies have found that patients’ beliefs about medications explain a significant part of the variation in medication nonadherence and refusal [40].

Participants who cited mild symptoms from prior COVID-19 vaccination were less resistant to receive a third booster COVID-19 vaccine dose. Furthermore, those categorized as having a low risk of developing COVID-19 complications were still hesitant to receive a third booster COVID-19 dose. This could be another manifestation of a benefit versus risk evaluation and the effect of perceived necessity and concerns about medications on therapy adherence [41]. Generally, it seems that beliefs about the COVID-19 pandemic and vaccines are evolving within populations. Previous work has recommended that communication campaigns and awareness activities be continuous and intense to maintain a perception of the severity of COVID-19, rather than the emerging popular belief that COVID-19 is no longer a threat [42,43].

A novel finding from the present study is how the severity of side effects of a particular vaccine influenced the decision to receive a third vaccine dose. Recipients of the AstraZeneca vaccine followed by Pfizer reported the most severe/moderate side effects. In turn, recipients of the Sinopharm vaccine reported mostly no or mild side effects. The reported side effects of the vaccines were concordant with what has been published elsewhere [27,36,44,45]. The severity of the vaccine side effects influenced the participants’ prospective decision to receive a third booster COVID-19 vaccine dose.

Furthermore, data clearly shows that false information about the COVID-19 vaccine is being spread online. Indeed, a previous study discovered that using social media to organize offline action is highly predictive of the belief that vaccinations are unsafe, with such beliefs increasing as more social media activity occurs. In addition, the prevalence of foreign disinformation is highly statistically and substantively significant in predicting a drop in mean vaccination coverage over time [19]. As a result, the WHO has increased its communication efforts to provide appropriate responses to swiftly growing misinformation transmitted via online media. People who have questions regarding the outbreak can use the WHO’s online search optimization to find credible sites [46]. Unfortunately, social media and other internet corporations do not successfully guide queries to reliable information; in fact, how to do so is a hotly debated topic. Nonetheless, while searching for information on the COVID-19 vaccine and other health-related issues, social media sites are starting to issue notifications or warnings that include links to reliable sources and fact-checkers [47], though the efficacy of such efforts to combat this problem is yet to be evaluated.

The current study also qualitatively explored the reasons for refusal of a third dose. The main reported reason for not receiving a third vaccine dose was the lack of solid scientific evidence about the advantage of the third dose, followed by the perceived lack of benefit from a third booster dose in the short term. To a lesser degree, participants reported that a mild experience of COVID-19 infection and the severity of side effects were also reasons for the likely refusal of a third booster COVID-19 vaccine dose.

More efforts should be exerted to increase vaccine acceptance among high risk and fragile populations, particularly healthcare workers. Strategies suggested to improve vaccination among this group include usage of mixed communication models, including circulating leaflets, using social media, and dedicated websites to promote vaccination among healthcare workers [16].

### Strengths and Limitations

The current study recruited a large sample which helped minimize any effects of bias. This study evaluated participants’ knowledge about COVID-19 and its available vaccines, protective practices against COVID-19, prior experience with COVID-19 vaccination, and intention to receive a booster dose. Due to the methodology of the study, which was based on convenience sampling and a self-completed online questionnaire, different biases may have influenced the results, including selectivity and recall biases. However, online surveys can provide a safe and private environment for the respondents to complete questionnaires honestly and accurately, thereby minimizing any possible social desirability or interviewer biases. Moreover, with continuous expansion of the Internet, both globally and within Jordan, participants recruited via online surveys will become increasingly representative of the target population [48,49]. This is evident by the similarities found between the characteristics of the study sample and the target population including sample distribution by governate, as, for example, the largest sample are resident in Amman followed by Zarqa and the north region (Irbid and Mafraq), which mirrors the Jordanian population distribution by governate [50]. Moreover, when excluding children under the age of 18, the largest age group among the Jordanian population is between 18 and 29, which represents 34% of the adult Jordanians, and the largest age group in the current study was between 18 and 29 (40%); similarities in other age groups were found as well. In the present study, nearly 70% of the study group was educated to a degree level or above; it was reported that the percentage of tertiary education in Jordan was 40% of the total population in 2014, and the percent is increasing annually [51]. Furthermore, the household average monthly income distribution is very comparable to that reported by the Jordanian Department of Statistics [52].

## 5. Conclusions

In conclusion, the findings of the current study revealed that there are evolving beliefs about the emerging third booster vaccine dose. These beliefs are expected to negatively influence the vaccination rate. Hence, further efforts regarding awareness campaigns and promoting the benefits and safety of the third COVID-19 vaccine dose are urged to maintain the desired effects of the global COVID-19 vaccination program.

## Figures and Tables

**Figure 1 vaccines-10-00410-f001:**
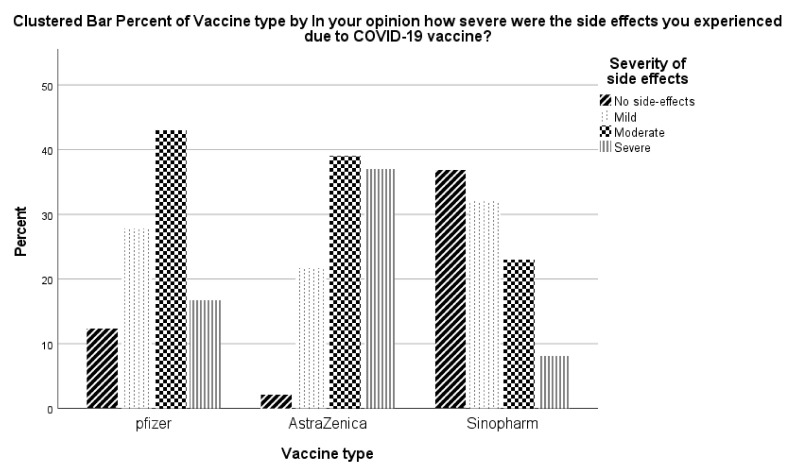
Severity of the side effects by vaccine type.

**Table 1 vaccines-10-00410-t001:** Sample characteristics.

		Frequency (%) (n = 915)
**Age**	18–29 years	418 (45.7)
	30–39 years	228 (24.9)
	40–49 years	143 (15.6)
	50–59 years	90 (9.8)
	60 years or more	36 (3.9)
**Marital status**	Married	457 (49.9)
**Do you have children?**	Yes	433 (47.3)
**Do you have a chronic disease?**	Yes	203 (22.2)
**Do you Smoke?**	No	660 (72.1)
	Ex-smoker	32 (3.5)
	Yes	223 (24.4)
**Educational level**	High school or less	66 (7.2)
	Diploma	41 (4.5)
	University student	168 (18.4)
	Bachelor	586 (64.0)
	Postgraduate	54 (5.9)
**Household average monthly income**	Less than 500 JD	340 (37.2)
	500–1000 JD	393 (43.0)
	More than 1000 JD	182 (19.9)
**Residence**	North region	87 (9.5)
	South region	23 (2.5)
	Zarqa	90 (9.8)
	Middle region	50 (5.5)
	Amman	631 (69.0)
	Other	34 (3.7)
**Side effects experienced from a COVID-19 vaccine**	No symptom	193 (21.1)
	Mild	266 (29.1)
	Moderate	323 (35.3)
	Severe	133 (14.5)
**Know someone who died due to COVID-19?**	Yes	677 (74.0)
**Reported side effects due to** **COVID-19 vaccine**	Headache	501 (54.8)
	Hyperthermia	517 (56.5)
	Pain at site of injection	615 (67.2)
	Muscle pain	275 (30.1)
	Weakness	365 (39.9)
	Spasm	356 (38.9)
	Nausea	158 (17.3)
	Rash	21 (2.3)
	Chills	192 (21.0)
	Water retention	22 (2.4)
	No symptoms	155 (16.9)

**Table 2 vaccines-10-00410-t002:** Intent to get the booster dose of the COVID-19 vaccine by different sample characteristics presented as medians (25–75 quartiles) or frequencies (percentages).

Variables	Are You Willing to Receive the Booster Dose for COVID-19 Vaccine			*p*-Value
	No (n = 281, 30.7%)	Not Sure (n = 226, 24.7%)	Yes (n = 408, 44.6%)	
**Participant Age**				0.167
18–29 years	138 (33.0)	111 (26.6)	169 (40.4)	
30–39 years	74 (32.5)	53 (23.2)	101 (44.3)	
40–49 years	39 (27.3)	36 (25.2)	68 (47.6)	
50–59 years	20 (22.2)	21 (23.3)	49 (54.4)	
60 years or more	10 (27.8)	5 (13.9)	21 (58.3	
**Do you have children?**				0.20
No	158 (32.8%)	122 (25.3%)	202 (41.9%)	
Yes	123 (28.4%)	104 (24.0%)	206 (47.6%)	
**Marital status**				0.20
Married	132(28.9%)	108 (23.6%)	217 (47.5%)	
Other	149 (32.5%)	118 (25.8%)	191 (41.7%)	
**Education level**				0.18
High school or less	17 (25.8)	12 (18.2)	37 (65.1)	
University student	59 (35.1)	47 (28.0)	62 (36.9)	
Diploma	17 (41.5)	7 (17.1)	17 (41.5)	
Bachelor’s degree	174 (29.7)	147 (25.1)	255 (45.2)	
Postgraduate	14 (25.9)	13 (24.1)	27 (50.0)	
**Household average monthly income**				0.10
Less than 500 JD	112 (32.9)	82 (24.1)	146 (42.9)	
500–1000 JD	119 (30.3)	108 (27.5)	166 (42.2)	
More than 1000 JD	50 (27.5)	36 (19.8)	96 (52.7)	
**In your opinion how severe were the symptoms you experienced due to COVID-19 vaccine?**				0.003
Mild	74 (27.8)	65 (24.4)	127 (47.7)	
Moderate	105 (32.5)	85 (26.3)	133 (41.2)	
Sever	58 (43.6)	28 (21.1)	47 (35.3)	
No symptom	44 (22.8)	48 (24.9)	101 (52.3)	
**Did you take the COVID-19 vaccine out of conviction, or you were forced by laws imposed by the state?**				<0.001
I took it out of conviction	52 (13.9)	58 (15.5)	264 (70.6)	
I took it out of conviction and because of the laws	45 (19.8)	92 (40.5)	90 (36.6)	
I took it because of the imposed laws, not because I believe in it	182 (59.1)	75 (24.4)	51 (16.6)	
**Do you know someone who has died from COVID-19?**				0.28
No	82 (34.7)	55 (23.3)	99 (41.9)	
Yes	198 (29.2)	171 (25.3)	308 (45.5)	
**In your opinion, how serious** **is COVID-19?**	3.5 (3–4)	3 (3–4)	4 (3–4)	0.03
**Risk level**				0.009
Low	139 (32.6)	123 (28.9)	164 (38.5)	
Medium	74 (28.1)	58 (22.1)	131 (49.8)	
High	68 (30.1)	45 (19.9)	113 (50.0)	
**Knowledge level**				0.09
Low	135 (33.7)	103 (25.7)	163 (40.6)	
High	146 (28.4)	123 (23.9)	245 (47.7)	
**Practice level**				0.23
Low	141 (32.9)	109 (25.4)	179 (41.7)	
High	140 (28.8)	117 (24.1)	229 (47.1)	
**Vaccine type**				<0.001
Pfizer	175 (33.3)	142 (27)	208 (39.6)	
AstraZeneca	21(45.7)	8 (17.4)	17 (37)	
Sinopharm	85(24.7)	76 (22.1)	183 (53.2)	

**Table 3 vaccines-10-00410-t003:** Multivariate predictors of responding “not sure” or “no” regarding intent to get the booster dose of the COVID-19 vaccine.

Characteristics	Intent to Vaccinate No vs. Yes OR (95%CI)	Intent to Vaccinate Not Sure vs. Yes OR (95%CI)
**In your opinion, how serious is COVID-19?**	0.91 (0.76–1.10)	0.90 (0.75–1.07)
**Participant Age**		
18–29 years	0.92 (0.31–2.79)	2.20 (0.65–7.49)
30–39 years	0.97 (0.34–2.81)	1.67 (0.51–5.50)
40–49 years	0.77 (0.26–2.32)	1.66 (0.49–5.59)
50–59 years	0.56 (0.18–1.79)	1.62 (0.47–5.60)
60 years or more	Reference	
**Marital status**		
Other	0.86 (0.39–1.92)	1.17 (0.51–2.68)
Married	Reference	
**Do you have children**		
No	1.25 (0.60–2.60)	0.73 (0.33–1.61)
Yes	Reference	
**Education level**		
High school or less	0.39 (0.13–1.17)	0.38 (0.21–1.10)
University student	0.95 (0.36–2.48)	0.70 (0.28–1.74)
Diploma	0.73 (0.23–2.32)	0.36 (0.10–1.23)
Bachelor’s degree	0.79 (0.34–1.84)	0.60 (0.27–1.35)
Postgraduate	Reference	
**Household average monthly income**		
Less than 500 JD	1.23 (0.72–2.10)	1.63 (0.95–2.78)
500–1000 JD	1.29 (0.77–2.17)	1.79 (1.07–3.0) *
More than 1000 JD	Reference	
**In your opinion how severe were the symptoms you experienced due to COVID-19 vaccine?**		
No symptom	0.34 (0.18–0.65) **	0.81 (0.43–1.54)
Mild	0.54 (0.31–0.97) *	0.87 (0.48–1.60)
Moderate	0.64 (0.37–1.13)	1.08 (0.60–1.96)
Severe	Reference	
**Did you take the COVID-19 vaccine out of conviction, or you were forced by laws imposed by the state?**		
I took it because of the imposed laws, not because I believe in it	20.88 (13.13–33.21) **	7.42 (4.58–12.01) **
I took it out of conviction and because of the laws	2.68 (1.66–4.34) **	4.99 (3.27–7.63) **
I took it out of conviction	Reference	
**Do you know someone who has died from COVID-19?**		
No	1.42 (0.93–2.17)	0.94 (0.61–1.44)
Yes	Reference	
**Knowledge level**		
Low	1.17 (0.81–1.70)	1.18 (0.82–1.71)
High	Reference	
**Risk level**		
Low	1.52 (0.94–2.46)	1.70 (1.06–2.74) *
Medium	0.89 (0.53–1.49)	0.93 (0.55–1.56)
High	Reference	
**Total practice score**		
Low	1.01 (0.69–1.47)	1.06 (0.73–1.52)
High	Reference	

* Significant at the level of less than 0.05, ** significant at the level of less than 0.01.

**Table 4 vaccines-10-00410-t004:** Reasons participants provided for responding “No” or “not sure” regarding intent to get a booster dose of the COVID-19 vaccine.

Reasons	Frequency (%)
The booster dose will not provide me with any further protection against COVID-19.	277 (30.3)
The booster dose will have severe side effects.	312 (34.1)
I can’t tolerate another dose because the side-effects of the previous ones were severe.	225 (24.6)
I was infected with COVID-19, therefore I do not need the booster dose.	120 (13.1)
The symptoms I experienced due to COVID-19 infection were mild; therefore, I will not receive the booster dose.	238 (26.0)
Taking the booster dose now has no benefit, however I may receive it in the future.	351 (38.4)
I took the last dose a short time ago, so there will be no need to take the booster dose for at least a year.	338 (36.9)
The benefits of a booster dose have not been scientifically proven.	364 (39.8)
The booster dose is a conspiracy to boost corporate profits.	318 (34.8)
A booster dose will be imposed only based on agreements reached between pharmaceutical companies and governments.	279 (30.5)

## Data Availability

The data presented in this study are openly available in [Zenodo] at [10.5281/zenodo.6026138], [53].

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
