# Peer review of "Willingness of the Jordanian Population to Receive a COVID-19 Booster Dose: A Cross-Sectional Study"

_vaccines, 2022, doi:10.3390/vaccines10030410_

Round 1

Reviewer 1 Report

The authors investigated the willingness of the Jordanian population to receive a COVID-19 vaccine booster dose via a cross-sectional study. I believe this is an important study given that most of the populations/ nations globally have received the initial two doses of the covid-19 vaccine. Now with the emergence the question is are people prepared to continue taking boosters to manage any new variants that may emerge. This is so because people's hesitancy of taking additional doses has increased because of the lack of solid scientific evidence about the advantage of taking a third dose. 

The manuscript is well written. The study involved a sample size (915) well over the estimated size (601) based on 95% significance level and 4% margin of error. The data was analysed using standard statistical applications and the results discussed in light of what is currently known globally.  The emerging beliefs are expected to negatively influence the vaccination rate not only in Jordan but in other parts of the world hence the work presented here is relevant.

Reviewer 2 Report

Thank you for asking me to review this article.

Investigating the determinants underlying the intention to vaccinate with a booster dose for COVID-19 vaccination is crucial in the current pandemic context. The public perception of a health risk plays a decisive role in influencing people's behaviour and is often subject to the influence of social, cultural and psychological determinants.

While overall the study is neat in its presentation it is felt that it cannot be published in its current form. In this regard, some of the most important observations are reported below:

In the introduction, the authors give ample space to the scientific evolution that led to the production and administration of anti-COVID-19 vaccines, however it is my belief that some aspects described in the literature as being of importance in the acceptance of the vaccine are overlooked: first among them the phenomenon of Vaccine Hesitancy. In fact, in the current pandemic, Vaccine Hesitancy is further discussed especially in relation to its influence on health decisions (e.g. acceptability to the administration of a pharmaceutical product) with particular reference to fragile population cohorts such as patients suffering from chronic degenerative diseases or particular cohorts of workers (e.g. healthcare workers). In my opinion, assessing this aspect is significant to the study and could add value to the research. As such, I recommend consulting the manuscripts DOI: 10.3390/vaccines9090971 and DOI: 10.1080/21645515.2020.1777821.

Furthermore, it does not contextualise the subject of the research in the era of web 2.0 and 3.0 in which the use of modern Information and Communication Techologies (ICT) is ever-increasing and modifying the way people communicate with each other. In particular, social media platforms have become one of the main sources of information for society, and thus, investigating the level of influence that these digital tools exert on society also with respect to people's choices and decisions can offer many insights and analysis. It is well reported in the literature that the decision-making process is strongly influenced by what the user reads online, especially in relation to certain widely discussed health issues e.g. vaccinations doi:   10.1136/bmjgh-2020-004206

Therefore, to explore this premise and in particular the influence that the perception of a health risk exerts on the adoption of the correct preventive measures with a view to implementing the best prevention strategies accompanied by adequate communication campaigns is significant and could increase the quality of the manuscript.

The methods are described in a discursive but not completely exhaustive manner; an outline or a table containing the questions or the six domains with which they were collected and, above all, the time frame in which the questionnaire was administered is required. At the time of the survey, what was the epidemiological situation related to the COVID-19 pandemic in Jordan? What was the national vaccination coverage of the young adult cohort targeted by the survey at the time the questionnaire was administered? How long had the baseline cycle (two doses) been completed in the population? This aspect could be included in the study setting with a precise outline that could better clarify these details, making for a smoother reading of the manuscript and opening up the discussion to other insights. Furthermore, in the description of the characteristics of the sample, there is a discrepancy between the absolute numbers per category and the total.

Regarding the description of the sample, the authors report a total of 915 users. In Table 1, under the section "Residence", the sum of the absolute number of patients amounts to 881 vs. the 915 declared. Therefore a verification of the data or the addition of any missing class is requested.

Furthermore, the legend in Table 1 is actually the concluding sentence of the previous paragraph. The authors are asked to rectify and check overall. Furthermore, the tables are not mentioned in the text and this does not make reading of the results fluent.

Line 143 "...A multinominal logistic regression with participants' intention to vaccinate their children as the dependent variable was conucted....." is not consistent with line 86 "....The dependent variable ('Are you willing to take the booster dose?')....". The authors are asked to clarify.

Line 114 '.........The second section gathered data on the individuals' assessment of COVID-19 disease severity on a scale of 1 (not serious) to 5 (extremely serious) and whether they were infected or thought they could be infected in the next 6 months.......'. The authors describe the second session of the questionnaire as resulting from a quantitative variable (on a scale of severity 1-5) and an undefined variable concerning the possibility of becoming infected. What is the resulting variable and how is it measured? This is unclear. Furthermore, in Tables 2 and 3, where the same variable is reported for the association "to willing to receive the booster dose..." as a multivariate result, respectively, the way in which the variable was input into the model is unclear.

In the "Discussion" section it is suggested to extend the empirical value of the results obtained in the context of the influence that the Web exerts on decisions to undergo vaccination also in the context of Vaccine Hesitancy for both the general population and healthcare workers as mentioned by the authors themselves in the introduction. 

There are also several typing errors in the text (i.e. COVID-419 in line 177) and typing errors in the tables (uninterpretable data). For example, in Table 3, section "In your opinion, how serious is COVID1b?", the data in brackets refer to what?

These discrepancies need to be resolved before a publication can be considered. By clarifying and justifying all these issues, in my opinion, the manuscript will be a useful contribution to scientific research.

Minor revisions: in relation to form, in order to improve the quality of the manuscript and make it more orderly, it is suggested that the same font and size be used for the text, legend and tables.

Author Response

Thank you for asking me to review this article.

Investigating the determinants underlying the intention to vaccinate with a booster dose for COVID-19 vaccination is crucial in the current pandemic context. The public perception of a health risk plays a decisive role in influencing people's behaviour and is often subject to the influence of social, cultural and psychological determinants.

While overall the study is neat in its presentation it is felt that it cannot be published in its current form. In this regard, some of the most important observations are reported below:

-Thank you for your comments.

In the introduction, the authors give ample space to the scientific evolution that led to the production and administration of anti-COVID-19 vaccines, however it is my belief that some aspects described in the literature as being of importance in the acceptance of the vaccine are overlooked: first among them the phenomenon of Vaccine Hesitancy. In fact, in the current pandemic, Vaccine Hesitancy is further discussed especially in relation to its influence on health decisions (e.g. acceptability to the administration of a pharmaceutical product) with particular reference to fragile population cohorts such as patients suffering from chronic degenerative diseases or particular cohorts of workers (e.g. healthcare workers). In my opinion, assessing this aspect is significant to the study and could add value to the research. As such, I recommend consulting the manuscripts DOI: 10.3390/vaccines9090971 and DOI: 10.1080/21645515.2020.1777821.

Furthermore, it does not contextualise the subject of the research in the era of web 2.0 and 3.0 in which the use of modern Information and Communication Techologies (ICT) is ever-increasing and modifying the way people communicate with each other. In particular, social media platforms have become one of the main sources of information for society, and thus, investigating the level of influence that these digital tools exert on society also with respect to people's choices and decisions can offer many insights and analysis. It is well reported in the literature that the decision-making process is strongly influenced by what the user reads online, especially in relation to certain widely discussed health issues e.g. vaccinations doi:   10.1136/bmjgh-2020-004206

Therefore, to explore this premise and in particular the influence that the perception of a health risk exerts on the adoption of the correct preventive measures with a view to implementing the best prevention strategies accompanied by adequate communication campaigns is significant and could increase the quality of the manuscript.

-Thank you for your comment. The following was added to the manuscript:

“Several studies have investigated the reasons behind vaccine hesitancy in general and influenza vaccine in different populations especially healthcare professionals. These studies have discovered that the main determinants of hesitation are 1) insufficient awareness campaigns; 2) altered risk perception; 3) insufficient health education on the efficacy of the influenza vaccine and/or potential adverse reactions; 4) lack of access to vaccination facilities; and 5) socio-demographic variables. Furthermore, according to various studies, one of the key factors of low flu vaccine uptake among healthcare workers is a lack of time to attend vaccination clinics[16]. Therefore, one of the suggested methods to combat vaccination hesitancy is an integrated vaccination offer[16]

A population's decision to get vaccinated is embedded in a specific social environment of ideas and perceptions, as well as concerns related to vaccine availability and costs. [17]The notion of vaccine hesitancy was proposed by the World Health Organization's Strategic Advisory Group of Experts with the goal of evaluating the social variables that lead to a delay in vaccine acceptance or rejection despite the availability of immunization services[18]. The issue of vaccine hesitancy has increased with the emergence of social media that has increased the spread of misconceptions associated with vaccination [19]”

The methods are described in a discursive but not completely exhaustive manner; an outline or a table containing the questions or the six domains with which they were collected and, above all, the time frame in which the questionnaire was administered is required. At the time of the survey, what was the epidemiological situation related to the COVID-19 pandemic in Jordan? What was the national vaccination coverage of the young adult cohort targeted by the survey at the time the questionnaire was administered? How long had the baseline cycle (two doses) been completed in the population? This aspect could be included in the study setting with a precise outline that could better clarify these details, making for a smoother reading of the manuscript and opening up the discussion to other insights.

-Thank you for your comment the following was added:

“The data were collected between 1 October and 15 December 2021. In that period, when the data were collected, the pandemic in Jordan rose from 700 daily new confirmed cases on the 1 October to 5000 daily new confirmed cases by 15 December [20]. At the beginning of data collection more than 3,290,000 adults had been completely vaccinated[21] which represents around 49% of the adult population of Jordan[22].”

Furthermore, Regarding the description of the sample, the authors report a total of 915 users. In Table 1, under the section "Residence", the sum of the absolute number of patients amounts to 881 vs. the 915 declared. Therefore a verification of the data or the addition of any missing class is requested.

-Thank you for your comment, the actual sample size is 915 and this is confirmed by the totals of all the other subgroups. The percentages previously included in the manuscript of the residency variable did not add up to 100%. The missing class (Other) was added which may refer to remote areas of Jordan (particularly around the Iraqi borders).

Furthermore, the legend in Table 1 is actually the concluding sentence of the previous paragraph. The authors are asked to rectify and check overall. Furthermore, the tables are not mentioned in the text and this does not make reading of the results fluent.

-There seems to be technical issues that affects the manuscript display. The manuscript is correctly displayed in the uploaded version (on my pc). Table 1 legend is “Sample Characteristics”. Reference to all the tables has been added to the text as suggested.

Line 143 "...A multinominal logistic regression with participants' intention to vaccinate their children as the dependent variable was conucted....." is not consistent with line 86 "....The dependent variable ('Are you willing to take the booster dose?')....". The authors are asked to clarify.

-We strongly apologize for this typo. The sentence was corrected to “A multinominal logistic regression with participants’ intention to receive the booster dose as the dependent variable”

Line 114 '.........The second section gathered data on the individuals' assessment of COVID-19 disease severity on a scale of 1 (not serious) to 5 (extremely serious) and whether they were infected or thought they could be infected in the next 6 months.......'. The authors describe the second session of the questionnaire as resulting from a quantitative variable (on a scale of severity 1-5) and an undefined variable concerning the possibility of becoming infected. What is the resulting variable and how is it measured? This is unclear. Furthermore, in Tables 2 and 3, where the same variable is reported for the association "to willing to receive the booster dose..." as a multivariate result, respectively, the way in which the variable was input into the model is unclear.

-The authors apologize for the vague sentence used in this section. The only question in this section that had a scale of 1 to 5 was the question about the seriousness of COVID-19 while the other questions had different response options, the complete translated questionnaire was included as an appendix for your evaluation. The severity question was the only question in this section that was include as a continuous variable, while other question had nominal type answers.

In the "Discussion" section it is suggested to extend the empirical value of the results obtained in the context of the influence that the Web exerts on decisions to undergo vaccination also in the context of Vaccine Hesitancy for both the general population and healthcare workers as mentioned by the authors themselves in the introduction. 

-Thank you for your comment. The following was added: Furthermore, data clearly shows that false information about the COVID-19 vaccine is being spread online. Indeed, a previous study discovered that using social media to organize offline action is highly predictive of the belief that vaccinations are unsafe, with such beliefs increasing as more social media activity occurs. In addition, the prevalence of foreign disinformation is highly statistically and substantively significant in predicting a drop in mean vaccination coverage over time [19]. As a result, the WHO has increased its communication efforts to provide appropriate responses to swiftly grow-ing misinformation transmitted via online media. People who have questions regarding the outbreak can use the WHO's online search optimization to find credible sites [47]. Unfortunately, social media and other internet corporations do not successfully guide queries to reliable information; in fact, how to do so is a hotly debated topic. Nonetheless, while searching for information on the COVID-19 vaccine and other health-related issues, social media sites are starting to issue notifications or "warnings" that include links to reliable sources and fact-checkers [48], though the efficacy of such efforts to combat this problem should be evaluated

More efforts should be exerted to increase vaccine acceptance among high risk and fragile population particularly healthcare workers. Strategies suggested to improve vaccination among this group include usage of mixed communication models including circulating leaflets, using social media, and dedicated websites to promote vaccination among healthcare workers [16]“ 

There are also several typing errors in the text (i.e. COVID-419 in line 177) and typing errors in the tables (uninterpretable data). For example, in Table 3, section "In your opinion, how serious is COVID1b?", the data in brackets refer to what?

-These typing errors were corrected, the data in the table are the mean rank used in Kruskal–Wallis analysis. However, the authors agree with the editor that it is hard to interpret, therefore, it was changed to medians (25-75 quartiles) and this was clarified in the table legends and descriptions 

These discrepancies need to be resolved before a publication can be considered. By clarifying and justifying all these issues, in my opinion, the manuscript will be a useful contribution to scientific research.

-Thank you for comments that significantly improved the quality of the manuscript. We hope that all your comments were managed to your satisfaction

Minor revisions: in relation to form, in order to improve the quality of the manuscript and make it more orderly, it is suggested that the same font and size be used for the text, legend and tables.

-Thank you for your comment changes were made accordingly

Reviewer 3 Report

This paper was easy to read with interesting results that add to the body of knowledge as to why people may be hesitant to take up a vaccine despite the scientific evidence that it will protect against a serious disease.  I have a general comment and some specific amendments.

There is always going to be some bias when selecting a population for questionnaire and the limitations are well described.  However I have a question that has not been addressed which is how representative of the overall Jordanian population was the study group? Although you suggest increasing access to online increases how representative they become you have not offered a comparison of the study group demographic with Jordanian population.  For example nearly 70% of the study group was educated to degree level or above.  How does that compare to the general population and could that level of education have affected the responses?    

Line 162-163 the text has become mixed up with the table header. Please amend. Also lines 156 - 162 are smaller (subheading size) font. 

Table 2 'sever' should read 'severe', also in the figure which also doesn't have a figure number.

Also Table 2 the line that reads 'In your opinion, how serious is...' says 1b which should be 19 and the numbers in brackets don't look right.

Are lines 179-182 supposed to be footnotes to Table 2 with smaller font or are they the main body of text? In which case increase font.

Round 2

Reviewer 2 Report

Dear Authors,

thank you for considering my suggestions. I believe the quality of the manuscript has significantly improved. I have only a minor comment the authors should consider before the manuscript is published.

In particular, in the Introduction section, the authors have added the sentences in lines 67-84. I suggest revising the first sentences by introducing the reason why influenza vaccination is an example in order to understand the vaccine hesitancy phenomenon. In fact, we know that before the COVID-19 vaccination, influenza represented the best well-known model among the vaccine-preventable epidemic/pandemic airborne diseases.

Author Response

Thank you for your comment the authors are glad that the reviewer was satisfied with the modifications made on the manuscript. As suggested by the reviewer the following sentence was added at the the beginning of the paragraph: "Influenza remains the best well-known model among the vaccine-preventable epidemic/pandemic airborne diseases prior to COVID-19. Therefore, influenza vaccination example can be used to understand the phenomenon of vaccine hesitancy"